# The Practical Impact of Bias against Minority Group Applicants in Resume Screening on Personnel Selection Outcomes

**Jisoo Ock**

Department of Business Administration, Pusan National University, Busan 46241, Korea; jisoo.ock@pusan.ac.kr

**Abstract:** Research has consistently shown that resume screening decisions, despite their practical importance and frequent use in practice, are prone to biases that disadvantage applicants in demographic minority groups. Using a two-stage multiple-hurdle selection simulation as an example (initial selection on resume scores, then selection on a composite of cognitive ability and conscientiousness test scores), the current study illustrates the practical impact that bias against ethnic minority group applicants in resume evaluation can have on the outcomes of selection. Results show that if the bias against minority group applicants creates even a modest level of deflation in the observed resume evaluation scores for minority group applicants, the selection rate for minority group applicants is expected to be meaningfully lower compared to the selection rate for majority group applicants, increasing the likelihood of adverse impact. These findings demonstrate in clear practical terms the critical importance of fair resume evaluations for improving the legal defensibility of selection. Going beyond the simple understanding that bias against minority group applicants in resume screening leads to lower diversity, the current study contributes to the previous literature by clearly outlining the expected effect that varying levels of discriminatory resume evaluation have on the practical outcomes of selection. Moreover, we illustrate these results under a realistic set of conditions implied from the personnel selection literature and meta-analyses of variables relevant to personnel selection.

**Keywords:** resume screening; hiring discrimination; adverse impact; personnel selection; Monte Carlo simulation

## 1. Introduction

Resume screening provides organizations an opportunity to quickly evaluate job applicants' credentials (e.g., educational background, previous work experience, extracurricular activities) and screen out under- or unqualified applicants in the initial stage of the selection process [1,2]. Many organizations use resume screening as a part of their selection process because it is a practically useful procedure that eliminates the need to expend too much cost, labor, and time for processing a high volume of irrelevant application traffic. This is especially meaningful to larger organizations that tend to attract a high volume of applications, which is now more prevalent with the advancement of online recruiting systems that provide organizations access to applicants around the world at a relatively low cost [3,4].

Although resume screening is useful for increasing the efficiency of the personnel selection process, research has also shown that resume evaluations and screening decisions are prone to unfair discrimination against demographic minority group applicants. For example, research has shown that resumes with ethnic minority cues (e.g., ethnic minority-sounding name, affiliation to ethnic minority social groups) have much higher odds of rejection than for resumes with ethnic majority cues (as much as four to six times) even when they are matched in terms of abilities, experience, and qualifications [5–9]. Research has also indicated that multiple minority status can lead to compounded disadvantageous effect on the likelihood of selection [6–10]. Reflecting this unfortunate reality, interviews

with college job applicants have shown that some recruiters and job incumbents encourage job applicants to conceal or downplay racial cues in resumes (such as changing distinctively ethnic minority sounding names, omitting involvement in ethnic minority social groups, known as resume whitening) to increase the likelihood that minority group applicants will receive a callback [11].

Discrimination against minority group applicants in resume screening (or any other parts of a selection process) can have severe negative consequences that threaten the long-term sustainability of an organization [12,13]. Namely, by unfairly screening out minority group applicants, organizations are significantly limiting the pool of human resources that are available to them by denying employment opportunities to qualified minority group applicants. Moreover, organizations that fail to offer equitable employment opportunities to all job applicants are potentially violating employment discrimination laws (e.g., Civil Rights Act of 1964) that can incur significant monetary (e.g., legal fees) and social (e.g., negative publicity from being involved in a litigation) costs for the organization. Just as importantly, discrimination against minority group applicants in resume screening is problematic for individual applicants, potentially denying them a fair chance for employment.

The current study examined the practical threat that biased resume screening decisions have on the diversity outcomes, legal defensibility of selection, and the quality of the selected workforce. Specifically, using a computer simulation that focuses on a realistic example of a specific selection scenario, we calculated the ratio of selection rate for minority group applicants over the selection rate for majority group applicants (called adverse impact ratio, which is explained later) to indicate the diversity outcomes of selection. These calculations were made across samples under varying specified levels of bias against minority group job applicants on resume evaluation scores that created mean observed score disadvantage for minority group applicants (assuming that the true mean resume evaluation scores for majority and minority group applicants are equal) and different selection ratios at different stages of selection in a multiple-hurdle selection. In addition, we calculated the level of criterion performance score for the selected applicants both at the overall level and for group-specific level (majority and minority group applicants). We also calculated the change in these values across varying levels of resume evaluation bias. Integrating different aspects of previous models of selection utility analysis [14,15] and simulation research on adverse impact [16–18], the current simulation demonstrates the practical impact that biased resume screening has on personnel selection outcomes.

In general, we know that bias against minority group applicants in resume evaluation will lead to lower diversity in the selected workforce. However, computer simulations have the advantage of going beyond this general understanding: the current simulation manipulated a set of realistic selection parameters that lead to precise estimates of expected selection outcomes across a number of independent variable conditions that are systematically varied, as well as the associated variance of these outcomes across samples. Estimation of selection outcomes and variation in these estimates across multiple conditions and samples turns out to be much more tractable through simulations. Specifically, we can precisely illustrate the consequences of resume evaluation bias on the practical outcomes of selection in more direct and specific terms that extend relevant empirical study findings in the literature.

*Adverse Impact Ratio*

The diversity outcome of selection was defined in terms of adverse impact. Adverse impact refers to differential selection rates within groups such that the selection rate for a protected group is disproportionately lower than the selection rate for another protected group. Specifically, there is evidence of adverse impact when a selection ratio for any protected group is less than four-fifths (or 80%) of the selection rate for the group with the highest selection ratio. For example, if an organization receives applications from 100 male applicants and 100 female applicants, and the organization hires 20 male applicants but

only 10 female applicants, there is evidence of adverse impact against female applicants because the hiring rate for females is less than four-fifths of the hiring rate for males (0.10/0.20 = 0.50). When adverse impact is found, it is said that there is prima facie evidence for discrimination in the selection procedure, in which case the organization needs to prove that the selection procedure it used is job-relevant (i.e., show evidence of criterion-related validity for job performance) to justify its use [19]. Although evidence of adverse impact does not automatically indicate violation of laws prohibiting discriminatory employment, it can potentially pose various undesirable legal and social costs that organizations would like to avoid. In the simulation, we calculated the ratio of the selection rate for minority group applicants over the selection rate for majority group applicants in each simulated selection. We called this adverse impact ratio. Higher adverse impact ratios indicate more equivalent selection rates between minority and majority group applicants, whereas lower adverse impact ratios indicate lower selection rate for minority group applicants relative to majority group applicants.

## 2. Materials and Methods

Simulations for the current study were based on a two-stage selection where applicants are screened first on resume evaluation scores (selection stage 1), then the subset of passing applicants are selected top-down on the unit-weighted composite of scores on a cognitive ability test and a conscientiousness measure (selection stage 2). Table 1 summarizes the input population correlation matrix and the predictor and criterion subgroup mean differences that were used to develop the simulations. The predictor intercorrelations, validities, and majority–minority group mean differences (expressed as standardized majority–minority group mean difference, $d$) for the two predictors (cognitive ability and conscientiousness; 0.72 and 0.06, respectively) and the criterion variable (overall job performance; 0.35) were borrowed from the meta-analytically derived mean estimates [20,21]. The criterion-related validity for resume evaluation scores was set at $r = 0.15$, borrowing from the population correlation estimates found between multiple fit perceptions (i.e., person–organization fit, person–environment fit, person–supervisor fit, person–group fit) and overall job performance [22]. The correlations between multiple fit perceptions and overall job performance were used to set the criterion-related validity for resume evaluation scores, based on the finding that the mechanism underlying the relationship between resume content evaluation and hiring recommendation often follows a clinical approach, where hiring recommendations are made based on hiring managers' subjective combination of multiple fit perceptions towards applicants that are subjectively inferred from resume contents [23].

**Table 1.** Effect sizes and intercorrelations between predictors and overall job performance.

| Variable | Intercorrelation Matrix | | | | |
|---|---|---|---|---|---|
| | $d$ | 1. | 2. | 3. | 4. |
| Predictors | | | | | |
| 1. Resume | 0.00 to 0.50 | - | | | |
| 2. Cognitive ability | 0.72 | 0.37 | - | | |
| 3. Conscientiousness | 0.06 | 0.51 | 0.03 | - | |
| Criterion | | | | | |
| 4. Overall job performance | 0.35 | 0.32 | 0.52 | 0.22 | - |

Note. The population intercorrelations, validities, and majority–minority group mean differences (expressed as standardized mean differences, $d$) for resume score, cognitive ability, and conscientiousness are from Roth et al. (2011) [21]. The majority–minority group mean difference for overall job performance is from McKay and McDaniel (2006) [20]. The majority–minority group mean difference values represent the population effect size between Blacks and Whites on the measurement scores for these variables.

Consistent with previous meta-analyses of employment research [24,25], the majority–minority proportion in the simulated applicant pool was set at 80% majority group and 20% minority group. The total number of applicants in each simulation was set at $N = 1000$. Additionally, it was assumed that the four study variables have a multivariate normal distribution with equivalent variance-covariance matrix but differing means between majority and minority applicant populations (see Table 2 for a summary of the simulation characteristics and parameters).

**Table 2.** Summary of the simulation characteristics and parameters.

| **Constant** | | |
|---|---|---|
| *Predictors* | | |
| Resume evaluation score, cognitive ability, conscientiousness | | |
| *Criterion* | | |
| Overall job performance | | |
| *Number of applicants* | | |
| $N = 1000$ | | |
| **Variable** | | |
| *Resume selection* | | |
| 0.20, 0.50, 0.80 | | |
| *Standardized observed mean difference on resume evaluation score between majority and minority groups due to biased resume evaluation* | | |
| $d$ = 0.00 to 0.50 in 0.10 increments | | |

### 2.1. Independent Variables

**Majority–Minority Group Difference on Resume Score.** The observed mean score difference between majority and minority group applicants on resume evaluation scores due to biased evaluation of minority group applicants' resume information was expressed as standardized mean difference ($d$). The mean differences varied between 0.00 (no bias) and 0.50 (high bias) in 0.10 increments.

**Selection Ratio.** Different organizations might employ different levels of selectivity at the resume screening stage depending on the specific needs of the organization. As mentioned, organizations may increase selection ratio at resume screening to provide an opportunity to be considered for employment to as many applicants as possible. However, organizations may screen out a sizable proportion of applicants at resume screening to reduce costs in terms of money, time, and labor associated with processing under- or unqualified applications. To reflect these different strategies that organizations might employ, we varied the selection ratio at the resume screening stage across a range of values reflecting low (0.20), medium (0.50), and high resume screening ratios (0.80), assuming that the overall proportion of applicants hired from the initial applicant pool is relatively low (net selection ratio = 0.20).

### 2.2. Dependent Variables

**Adverse Impact Ratio.** For each simulated selection, we calculated the adverse impact (AI) ratio, defined as the ratio of the selection rate for minority group applicants over the selection rate for majority group applicants. We calculated both stage-specific and cumulative AI ratios.

**Overall Job Performance.** In addition to achieving a diverse workforce through selection, the effectiveness of personnel selection is also informed by the degree to which it serves to increase the overall job performance of the workforce. To reflect this aspect of selection, we examined the effect that the independent variables have on the expected overall job performance of the selected applicants. We calculated the expected overall job performance score at each selection stage and separately for the majority and minority group applicants.

*2.3. Procedures*

Sample realizations of measurement scores for the three predictors and the overall job performance criterion were generated based on the specified population correlation matrix in Table 1 (see [26], for the singular value decomposition method employed). Then, the mean majority–minority group differences on the predictors and the overall job performance criterion were imposed according to the meta-analytically derived standardized mean difference values described above. After generating the observed scores for the predictors and the criterion, the simulation performed top-down selection with the two-hurdle selection procedure that vary across the independent variables mentioned above. With each combination of the simulation parameters, the aforementioned process was replicated 1000 times to model sampling error variance within each condition. Simulations were programmed using R Code [27]. The codes used to generate the simulations are available for download online https://osf.io/uej7k/?view_only=d8edf6ea3da6408e9a524ab0d0fbf15c (accessed on 1 July 2022).

**3. Results**

The simulation results are presented in Table 3. Results showed that although bias against minority group applicants on resume evaluation scores was associated with lower AI ratio, the AI ratio at resume screening (selection stage 1) remained relatively high with a higher selection ratio at resume screening. For example, when the selection ratio at resume screening was as high as 0.80, the overall mean AI ratio remained high (above 0.80) even when the bias against minority applicants on the resume scores was as high as $d = 0.50$. However, higher selection ratio at resume screening was also associated with lower cumulative AI ratio. Within each resume score bias condition, AI ratio at selection stage 2 decreased as resume selection ratio increased. However, as the bias against minority applicants on resume scores increased above $d = 0.20$, the final AI ratio remained very low, regardless of the level of selection ratio at resume screening (see Figure 1). An expanded summary of the results that includes the interquartile range of the AI ratio estimates are available online https://osf.io/uej7k/?view_only=d8edf6ea3da6408e9a524ab0d0fbf15c (accessed on 1 July 2022).

**Table 3.** Stage-specific and cumulative mean adverse impact ratio.

| Net SR | Stage | SR | $d = 0.00$ | $d = 0.10$ | $d = 0.20$ | $d = 0.30$ | $d = 0.40$ | $d = 0.50$ |
|---|---|---|---|---|---|---|---|---|
| | S1 | 0.20 | 1.01 | 0.89 | 0.77 | 0.65 | 0.56 | 0.47 |
| | S2 | 0.50 | 0.56 | 0.59 | 0.63 | 0.65 | 0.70 | 0.72 |
| | Net | 0.10 | 0.57 | 0.53 | 0.49 | 0.43 | 0.40 | 0.35 |
| 0.10 | S1 | 0.50 | 1.00 | 0.93 | 0.85 | 0.77 | 0.71 | 0.64 |
| | S2 | 0.20 | 0.40 | 0.43 | 0.45 | 0.49 | 0.53 | 0.57 |
| | Net | 0.10 | 0.40 | 0.40 | 0.38 | 0.38 | 0.37 | 0.36 |
| | S1 | 0.80 | 1.00 | 0.96 | 0.93 | 0.89 | 0.85 | 0.81 |
| | S2 | 0.13 | 0.37 | 0.39 | 0.40 | 0.41 | 0.43 | 0.45 |
| | Net | 0.10 | 0.37 | 0.38 | 0.37 | 0.37 | 0.36 | 0.36 |
| **Net SR** | **Stage** | **SR** | **$d = 0.00$** | **$d = 0.10$** | **$d = 0.20$** | **$d = 0.30$** | **$d = 0.40$** | **$d = 0.50$** |
| | S1 | 0.50 | 1.00 | 0.97 | 0.85 | 0.78 | 0.70 | 0.64 |
| | S2 | 0.60 | 0.62 | 0.65 | 0.67 | 0.70 | 0.73 | 0.76 |
| 0.30 | Net | 0.30 | 0.63 | 0.60 | 0.57 | 0.55 | 0.51 | 0.49 |
| | S1 | 0.80 | 1.00 | 0.97 | 0.93 | 0.89 | 0.85 | 0.81 |
| | S2 | 0.38 | 0.52 | 0.53 | 0.55 | 0.57 | 0.59 | 0.61 |
| | Net | 0.30 | 0.52 | 0.51 | 0.51 | 0.51 | 0.50 | 0.50 |
| **Net SR** | **Stage** | **SR** | **$d = 0.00$** | **$d = 0.10$** | **$d = 0.20$** | **$d = 0.30$** | **$d = 0.40$** | **$d = 0.50$** |
| | S1 | 0.80 | 1.00 | 0.97 | 0.93 | 0.89 | 0.85 | 0.82 |
| 0.50 | S2 | 0.63 | 0.66 | 0.67 | 0.69 | 0.70 | 0.72 | 0.74 |
| | Net | 0.50 | 0.66 | 0.65 | 0.64 | 0.63 | 0.62 | 0.60 |

Note. S1 = selection stage 1 (resume screening); S2 = selection stage 2 (selection on unit-weighted composite of cognitive ability and conscientiousness test scores); Net = cumulative AI ratio; $d$ = standardized observed mean difference on resume evaluation score between majority and minority group applicants due to biased resume evaluation.

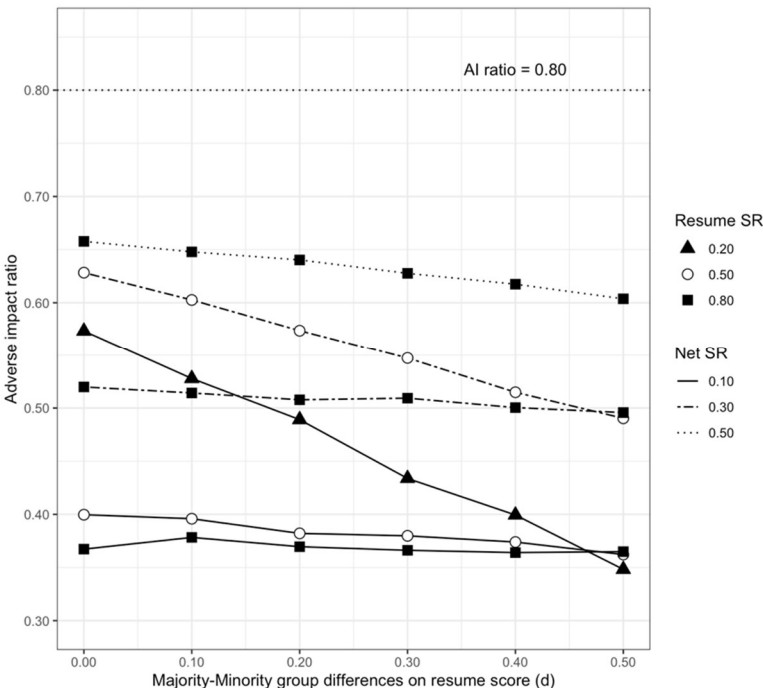

**Figure 1.** Change in mean cumulative adverse impact ratio associated with change in majority–minority group mean differences on resume evaluation score across selection ratio conditions.

Previous empirical studies have shown that discrimination against minority group applicants can meaningfully undermine minority group applicants' likelihood of getting the opportunity to be considered further for a job [5–9]. The current simulation results extend the previous literature by showing that even a modest level of bias against minority group applicants in resume evaluation is expected to have a severe discriminatory effect on selection of minority group applicants, regardless of any adjustments to the selection procedures (e.g., selection ratios) that organizations might make to improve the diversity outcomes of selection.

In addition to adverse impact, we examined the effect that the study variables have on the overall job performance of the selected applicants. The group-specific mean overall performance scores for the selected applicants are presented in Table 4. Results show that the level of bias against minority group applicants on resume evaluation scores had minimal effect on the mean criterion performance of the selected applicants, both at the group-specific level and at the overall level. Specifically, across the selection ratio conditions, the largest difference in mean criterion performance between the no bias condition (i.e., $d = 0.00$) and high bias condition (i.e., $d = 0.50$) was 0.01 for majority group applicants and 0.04 for minority group applicants (net SR = 0.10 and resume SR = 0.20 for both groups).

Table 4 also shows that the higher selection ratio at resume screening was associated with higher mean criterion performance in the selected applicants for both majority and minority group applicants. However, the mean level of increase was generally small. For example, in the low net selection ratio (net SR = 0.10), the difference in mean criterion performance between applicants selected in the low resume selection ratio condition (resume SR = 0.20) and applicants selected in the high resume selection ratio condition (resume SR = 0.80) was $d = 0.12$ (0.87–0.75) for majority group applicants and $d = 0.15$ (0.75–0.60) for minority group applicants. An expanded summary of the results that includes the interquartile range of the overall job performance estimates are available online https://osf.io/uej7k/?view_only=d8edf6ea3da6408e9a524ab0d0fbf15c (accessed on 1 July 2022).

**Table 4.** Group-specific (majority and minority group) mean overall job performance scores for selected applicants.

| Net SR | Resume SR | d = 0.00 | | d = 0.10 | | d = 0.20 | | d = 0.30 | | d = 0.40 | | d = 0.50 | |
|---|---|---|---|---|---|---|---|---|---|---|---|---|---|
| | | Maj | Min | Maj | Min | Maj | Min | Maj | Min | Maj | Min | Maj | Min |
| 0.10 | 0.20 | 0.75 | 0.60 | 0.75 | 0.60 | 0.76 | 0.60 | 0.75 | 0.62 | 0.75 | 0.62 | 0.75 | 0.64 |
| | 0.50 | 0.86 | 0.73 | 0.85 | 0.73 | 0.85 | 0.73 | 0.85 | 0.74 | 0.85 | 0.72 | 0.85 | 0.73 |
| | 0.80 | 0.87 | 0.75 | 0.87 | 0.74 | 0.87 | 0.77 | 0.87 | 0.74 | 0.86 | 0.75 | 0.87 | 0.75 |
| **Net SR** | **Resume SR** | **d = 0.00** | | **d = 0.10** | | **d = 0.20** | | **d = 0.30** | | **d = 0.40** | | **d = 0.50** | |
| | | Maj | Min | Maj | Min | Maj | Min | Maj | Min | Maj | Min | Maj | Min |
| 0.30 | 0.50 | 0.51 | 0.33 | 0.51 | 0.33 | 0.51 | 0.34 | 0.51 | 0.36 | 0.51 | 0.35 | 0.51 | 0.37 |
| | 0.80 | 0.56 | 0.41 | 0.55 | 0.42 | 0.56 | 0.42 | 0.55 | 0.41 | 0.56 | 0.42 | 0.56 | 0.42 |
| **Net SR** | **Resume SR** | **d = 0.00** | | **d = 0.10** | | **d = 0.20** | | **d = 0.30** | | **d = 0.40** | | **d = 0.50** | |
| | | Maj | Min | Maj | Min | Maj | Min | Maj | Min | Maj | Min | Maj | Min |
| 0.50 | 0.80 | 0.36 | 0.19 | 0.36 | 0.19 | 0.37 | 0.19 | 0.36 | 0.19 | 0.36 | 0.20 | 0.36 | 0.19 |

Note. Net SR = cumulative selection ratio; resume SR = selection ratio at resume screening stage; *d* = standardized observed mean difference on resume evaluation score between majority and minority group applicants due to biased resume evaluation.

## 4. Discussion

As the workforce in the US becomes increasingly diverse in terms of ethnicity, race, and gender [28], there is a strong need for research that helps organizations to effectively identify and hire talented employees from a larger and more diverse pool of applicants. Given the consistent stream of evidence indicating discrimination against minority group job applicants in resume screening, the current study used a computer simulation to illustrate the practical threat that unfair discrimination against minority group applicants at resume screening can have on the practical outcomes of selection, and whether increasing the selection ratio at the resume screening could effectively help organizations achieve more workforce diversity. Going beyond the simple understanding that bias against minority group members on resume evaluation leads to more discriminatory selection outcomes, the current simulations incorporated the effect of selection ratio at the resume screening stage and at the overall level under a realistic set of conditions implied from the personnel selection literature and the meta-analysis of variables relevant to personnel selection. Additionally, we examined the tradeoffs between adverse impact and validity, as well as how much the selection outcomes are expected to vary across multiple samples.

One of the main findings of the current study was that increasing the selection ratio at the resume screening stage, thus mitigating discrimination against minority group applicants at resume screening and allowing more applicants to be considered for employment, could serve to decrease the expected level of minority applicant hiring rate in the aggregate. This seemingly contradictory pattern of results occurred because with a given net selection ratio, a higher selection ratio at the initial stage of selection led to a lower selection ratio at the second stage and placed more weight on the predictor scores at this stage. Then, because of the substantial subgroup mean differences on the cognitive ability test score, minority group applicants were less likely to be selected at the second stage of selection, even though the subgroup mean differences on the cognitive ability test score was somewhat compensated by the smaller subgroup mean differences on the conscientiousness score that was used to form the unit-weighted composite in the second stage of the selection. Consequently, higher AI ratio at resume screening (due to high selection ratio at resume screening) was associated with lower cumulative AI ratio.

These results suggest that when the resume evaluation scores are valid and fair (as one would hope), adjusting the level of selectivity at different stages of the selection process in multiple-hurdle selection can be an effective method of reducing the likelihood of unfair discrimination against minority group applicants in selection. Namely, organizations can expect to increase diversity in the selected subgroup by being more selective (i.e., lower selection ratio) at the resume screening stage and imposing less selective selection ratios at the subsequent selection stages that involve predictor measures that show a large

majority–minority group mean differences. These results are consistent with previous research that has shown that when initial selection decisions are made on a valid predictor measure with a large majority–minority group mean difference (i.e., cognitive ability), then final selection decisions are made on an equally valid predictor measure with a smaller majority–minority group mean difference, organizations should be less selective at the initial selection stage in order to achieve greater diversity [16]. That being said, if the bias against minority group applicants in resume screening stage creates even a modest level of decrease in the observed mean resume evaluation scores for minority group applicants (i.e., $d > 0.20$), adjusting the selection ratio at resume screening had little impact on improving the diversity outcomes of selection.

The current study results also have important practical implications for assessing diversity outcomes in personnel selection. Namely, the results suggest that bias against minority group applicants in the initial selection stage can lead to underestimation of minority group hiring rate. That is, if organizations use some "quick-and-dirty" procedures to screen applicants prior to selection, and then calculate AI ratio only based on applicants who pass the pre-screening procedure, the AI ratio is likely to be overestimated (and the level of overestimation is expected to be higher to the extent that prior selection is more biased and selective). Then, to the extent that minority group applicants are prevented from even entering the selection process because of a biased pre-screening selection process, the actual level of minority group applicant hiring rate in the aggregate is expected to be much lower than when minority group applicant hiring rate is calculated only for applicants who have been considered for selection. This means that without careful consideration of the level of discrimination in pre-screening, calculations of minority applicant hiring rate may provide very misleading information about the fairness of selection.

Another important contribution of the current study was illustrating the tradeoffs that occurred between diversity and the expected level of criterion performance as selection ratios were adjusted to put more (or less) weight on the composite of cognitive ability and conscientiousness test scores (i.e., selection stage 2). Specifically, although lower selection ratio at resume screening and higher selection ratio at the second selection stage was associated with some loss in the expected level of criterion performance, this was also associated with higher AI ratios (given that the bias against minority group applicants on resume scores was small). The relatively small loss in the level of expected overall job performance might be acceptable to organizations, given that it is accompanied by a higher expected rate of selection of minority group applicants that meaningfully decreases the likelihood of litigation and negative publicity from alleged discrimination against minority group applicants in selection.

Finally, our findings imply that organizations should make concentrated efforts to minimize the level of bias against minority group applicants in resume screening. For example, instead of making resume screening decisions with human raters, who are prone to job-irrelevant biases that can undermine the fairness of those decisions, computers (i.e., artificial intelligence) can be trained to make resume screening decisions by identifying keywords that suggest that the applicant is qualified for the job [29,30]. Although computers are certainly also fallible to discriminatory selection decisions [31], they hold strong potential as a tool for improving the efficiency and fairness of resume screening decisions. This is especially true today as calls for collaborative works between computer scientists (who can develop the artificial intelligence-driven tools for resume screening) and HR experts (who hold expertise in personnel selection) continue [32], with the hope that such collaborative work will lead to development of artificial intelligence-driven selection tools that account for the need to select qualified and diverse workforce. Moreover, we argue that the effort to minimize bias against minority group applicants in resume screening should be a part of a greater effort to promote diversity in the workplace and to effectively manage workforce diversity, all of which are expected to contribute to the competitive advantage and long-term sustainability of an organization [12,13].

*Limitations and Future Research Directions*

Some limitations should be taken into consideration when interpreting the results of the current study. First, although our input correlation matrix was based on current meta-analytic estimates of predictor intercorrelations and criterion-related validity, these estimates may meaningfully deviate under specific situations in ways that are not due to sampling error. For example, different recruiters can rely on different resume information to develop employability judgments about different job applicants, which means that the validity of resume evaluation scores is expected to vary depending on how recruiters use the resume information to assess job applicants. Previous research has shown that recruiters make multiple inferences about job applicants' abilities and attributes based on their resume information (e.g., cognitive ability, personality, multiple fit perceptions [23,33,34]). However, if different recruiters have idiosyncratic perceptions about the importance of different abilities and individual attributes that are assessed in resume evaluation for predicting effective job performance (as is typically the case), the validity of these inferences should vary depending on the weights that are placed in combining the multiple inferences into making hiring recommendations, not to mention the expected unreliability of inferences that different recruiters make about job applicants based on resume information. Future research could investigate how the current study results may change depending on the level of criterion-related validity and reliability of resume evaluation scores (or other pre-screening methods that vary in their effectiveness for predicting job performance).

Second, the assumptions and scenarios considered in the current simulations do not reflect the wider range of possible selection parameters and conditions. Thus, rather than viewing the specific simulation results of the current study as a definitive summary, they should be viewed as an illustration of the general trends and principles of the effect that bias in resume screening and selection ratios could have across a range of selection situations for a range of different jobs. Future simulations could certainly extend the boundaries of our simulations by exploring additional parameters that were not modeled in the current study and their effect on the practical outcomes of selection.

Third, more research is needed to understand how organizations make resume screening decisions in practice and the psychometric properties of resume evaluation scores. Previous research has shown that recruiters use resumes to assess job applicants' academic qualifications, work experience, and extracurricular activities, which independently and interactively predict recruiters' employability ratings [33]. However, the effect of resume information on employability ratings has been shown to be somewhat complex. For example, Cole and colleagues [35] found that applicants with resumes depicting high academic qualifications were rated by recruiters as being highly employable, but applicants with resumes depicting low academic qualifications and high work experience were regarded as more employable than applicants with resumes depicting high academic qualifications and low work experience. Research has also shown that resume scores are related to the applicants' cognitive ability and personality [1], providing evidence that resume scores capture some information about psychological constructs that underlie effective job performance. However, it is not clear whether standardized resume evaluation processes are common in practice. Instead, recruiters might rely on some idiosyncratic intuitions in making resume screening decisions. This is problematic because studies have consistently shown that high-stakes decisions that are made based on clinical judgments of subjective information are less accurate compared to high-stakes decisions that are based on actuarial judgments using measurements that provide reliable and valid measurement scores [36,37]. Future research that identifies the different sources of errors in resume screening (due to idiosyncrasies in how organizations make resume screening decisions) and quantitatively model those errors should provide important extensions to the personnel selection literature.

## 5. Conclusions

Although resume screening is useful for improving the efficiency of personnel selection, research has shown that resume screening decisions tend to discriminate against demographic minority group applicants. The current study results showed that adjusting the selection ratio at the resume screening stage to place less weight on predictor tests that have large majority–minority group mean differences (i.e., cognitive ability) can meaningfully improve the rate at which minority group applicants are selected, given that resume evaluation scores are valid and fair. However, when resume scores are even modestly biased against minority group applicants, the overall selection ratio or selection ratio at different stages of selection had little impact on diversity outcomes of selection. Thus, fair resume evaluation is essential for achieving improved diversity outcomes of selection. Moreover, increasing the selection ratio at resume screening stage to allow more minority group applicants to be considered for employment can have an unintended consequence of decreasing the diversity outcomes of selection in the aggregate. These results illustrate that organizations cannot increase workforce diversity simply by considering more minority group applicants as job candidates. Rather, there is a need to place a sustained effort on identifying more valid and reliable methods of selection that do not unfairly penalize minority group applicants.

**Funding:** This work was supported by a 2-Year Research Grant from Pusan National University.

**Institutional Review Board Statement:** Not applicable.

**Informed Consent Statement:** Not applicable.

**Data Availability Statement:** Computer codes used to generate the simulations are available for download online (https://osf.io/uej7k/?view_only=d8edf6ea3da6408e9a524ab0d0fbf15c, accessed on 1 July 2022).

**Conflicts of Interest:** The author declares no conflict of interest. The funders had no role in the design of the study; in the collection, analyses, or interpretation of data; in the writing of the manuscript; or in the decision to publish the results.

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
