# Peer review of "The Practical Impact of Bias against Minority Group Applicants in Resume Screening on Personnel Selection Outcomes"

_sustainability, doi:10.3390/su14159438_

Round 1

Reviewer 1 Report

Explain the Monte Carlo simulation and indicate the associated references/literature

A research question/problem is missing. Introduce the research problem and place the hypotheses set.

Author Response

Please see the attachment for responses to comments by Reviewer 1. 

Reviewer 2 Report

The current manuscript is written and presented with details in the research steps and results. Some minor points are required to improve or clarify.

 1.There is no research hypotheses constructed and empirically tested in this paper. It will be more rigorous if research hypotheses are constructed from theories and/or existing literature.

2. In the abstract, try to improve the summary of results and the implications.

3. I would have liked to see how the empirical analysis and the key empirical findings presented in this paper are related to findings from previous papers (you should do this in the results section). This would add more value to the analyses and make it more susceptible to citations.

4.There are no references to the Sustainability journal. This makes me think that the paper is not necessarily related to the scope of the journal.

5. The cited references are not recent publications.

Author Response

Please see the attachment for responses to comments made by Reviewer 2. 

Reviewer 3 Report

Journal: Sustainability (ISSN 2071-1050)

Manuscript ID: Sustainability-1825772

Manuscript Type: Article

Title: The Practical Impact of Bias Against Minority Group Applicants in Resume Screening on Personnel Selection Outcomes

Review Report

Using a Monte Carlo simulation, this paper investigates 1) the practical impact that biases against minority group applicants at resume screening may have on the diversity outcomes of selection and 2) whether or not increasing the selection ratio at the resume screening could effectively help organizations achieve more diversity at work. It documents 1) that if the practical threat against minority group applicants in the resume screening stage creates even a modest level of difference in the observed mean resume evaluation scores between majority and minority group applicants, diversity of the selected workforce is more likely to be uniformly lower, and 2) higher AI ratio at resume screening, due to high selection ratio at resume screening, is associated with lower cumulative AI ratio. Overall, I find that it is an interesting research project and it is also a very concurrent subject matter we should care about. The reported results are mostly consistent with earlier studies. My only concern is that it does not consider the fact that there are very many other published papers in this area, e.g., Derous and Ryan, 2019. If possible, consider expanding the literature review so that you can locate the incremental contribution to the existing literature. I feel that some clear explanations of the tangible contributions to the literature are missing. I wish you the very best in publishing this manuscript. 

Author Response

Please see the attachment for responses to comments made by Reviewer 3. 
